# Impact on Prevalence of the Application of NAFLD/MAFLD Criteria in Overweight and Normal Weight Patients

**DOI:** 10.3390/ijerph191912221

**Published:** 2022-09-27

**Authors:** Ana Luisa Ordoñez-Vázquez, Eva Juárez-Hernández, Julia María Zuarth-Vázquez, Martha Helena Ramos-Ostos, Misael Uribe, Graciela Castro-Narro, Iván López-Méndez

**Affiliations:** 1Gastroenterology and Obesity Unit, Medica Sur Clinic & Foundation, Mexico City 14050, Mexico; 2Translational Research Unit, Medica Sur Clinic & Foundation, Mexico City 14050, Mexico; 3Internal Medicine Unit, Medica Sur Clinic & Foundation, Mexico City 14050, Mexico; 4Diagnosis and Treatment Unit, Medica Sur Clinic & Foundation, Mexico City 14050, Mexico; 5Transplants and Hepatology Unit, Medica Sur Clinic & Foundation, Mexico City 14050, Mexico

**Keywords:** metabolic syndrome, body mass index, liver disease, NAFLD, MAFLD

## Abstract

Background: Non-alcoholic fatty liver disease (NAFLD) is considered the hepatic manifestation of metabolic syndrome. Recently, the term metabolic dysfunction-associated fatty liver disease (MAFLD) has been proposed and adapted to body mass index (BMI). This study describes the impact on prevalence of the application of both criteria in overweight and lean patients. Methods: Patients who were evaluated for liver steatosis by transient elastography were included and divided according to BMI (≥25 kg/m^2^ and <25 kg/m^2^) and classified as NAFLD or MAFLD, according to metabolic abnormalities. Differences in prevalence were evaluated applying both criteria. A multivariate analysis was performed to evaluate independent associations of metabolic abnormalities and liver steatosis in lean patients. Results: 3847 patients were included. In overweight patients (61%), the prevalence NAFLD was 63.6% and 65.3% for MAFLD (*p* = 0.22). In contrast, the prevalence of MAFLD was lower (7.9% vs. 18.3%, *p* ≤ 0.001) in lean patients. In this group, higher age, fasting glucose, triglycerides, and waist circumference showed independent association with liver steatosis. Conclusion: The application of NAFLD/MAFLD criteria did not show prevalence differences in overweight patients. With MAFLD criteria, the prevalence is lower in lean patients, but patients with high risk of progression of liver disease for steatosis were identified, according to their metabolic abnormalities.

## 1. Introduction

Non-alcoholic fatty liver disease (NAFLD) is the most common liver disease in Western countries, with an estimated prevalence of 17–36% in adults. Differences are attributed to age, ethnicity, sex, and diagnostic method. It is a slowly progressive disease considering the hepatic manifestation of metabolic syndrome that could progress to steatohepatitis, fibrosis, cirrhosis, and hepatocellular carcinoma. A high-calorie diet, excess of refined carbohydrates, sugar-sweetened beverages, and high fructose intake have all been associated with NAFLD [1].

There are different hypotheses related to liver steatosis genesis. The most elucidated are the mechanisms related to obesity, insulin resistance, triglycerides, and fatty acid metabolism. The development of steatosis, as well as the progression to steatohepatitis and fibrosis, represents a complex and dynamic process. The combination of risk factors, host genetics, and gut microbiota can lead to an excessive influx of free fatty acids and accumulation of triglycerides in the hepatocytes, creating a lipotoxic environment that leads to liver inflammation, fibrosis, hepatocyte cell death, and pathological angiogenesis. The subsequent inflammatory response promotes fibrogenesis in the liver and is an important driving force for disease progression [2,3].

Despite these mechanisms being important, it seems to be that these are mediated by other more complex mechanisms, in which the gut–liver axis, bile acids metabolism, gut microbiota, endogenous ethanol, and intestinal catabolism of fructose are involved. Recently, several studies have been associated dysbiosis and loss of commensal bacterial metabolic functions with NAFLD, indicating that there are different mechanisms by which gut microbiota can contribute to NAFLD: intestinal inflammation, gut permeability dysfunction, energy intake and anaerobic fermentation, energy homeostasis, and bile acid metabolism [4,5,6,7].

Nowadays, due to the lack of an effective pharmacological treatment, lifestyle modifications, including diet and physical activity, are the first-line treatment options for NAFLD [8]. The effect of different diets has been studied in NAFLD patients, but due to the biological mechanisms of the Mediterranean diet, such as anti-inflammatory and antioxidant effects, as well as lipid lowering effect and modulation of gut microbiota through short fatty acid production, it has been proposed as an ideal diet for NAFLD patients [9,10].

In 2020, a panel of experts published a consensus statement to re-name NAFLD as metabolic dysfunction-associated fatty liver disease (MAFLD). Traditionally, NAFLD was considered the hepatic manifestation of metabolic syndrome; however, liver steatosis is recognized as a standalone disease characterized by a state of systemic metabolic dysfunction with MAFLD criteria [11]. The new proposal highlights insulin resistance and metabolic disorders as the major underlying causes of liver steatosis and allows to identify a group of patients with higher cardiovascular risk and liver complications [12].

Although liver steatosis is associated with overweight and obesity, it also occurs in lean subjects. The prevalence has been estimated in 3 to 7% in the general population; the differences could be explained by geographical regions, genetic alterations, and cut-off points of body mass index (BMI). These patients present similar a cardiovascular risk to overweight patients; nonetheless, pathophysiology of liver steatosis in lean subjects has not been elucidated, and factors such as diet, ethnicity, and gut microbiome have been proposed as triggers [13]. In these patients, besides metabolic factors, other mechanisms have been related to development NAFLD/MAFLD such as higher serum levels of secondary bile acids and fibroblast growth factor-19, as well as impairments in gut microbiota profile [14]. Regarding the progression of liver disease in lean patients, a recent meta-analysis shows that 39% had steatohepatitis, 29.2% had fibrosis, and 3% presented cirrhosis with an incidence of liver related mortality of 4.1 per 1000 persons per year [15]. The new proposed definition includes lean patients, denominated lean MAFLD, in whom metabolic abnormalities are taken into account for diagnosis, besides evidence of liver steatosis [11].

The new definition of MAFLD does not significantly change the prevalence compared with NAFLD, but it seems to reduce the incidence by 25% [16]; little is known about the epidemiologic changes among lean patients applying the MAFLD criteria and the differences in metabolic profiles. The aim of this study is to describe changes in prevalence and metabolic profiles using both definitions for this disease in overweight and lean patients.

## 2. Materials and Methods

### 2.1. Patient Population

The population of this retrospective study was selected from a series of consecutive patients who attended a check-up unit of Medica Sur Clinic & Foundation between 2019 and 2020. Inclusion criteria were Hispanic patients of both genders, older than 18 years old. The exclusion criteria included alcohol intakes > 2 drinks per day in women and >3 drinks per day in men, known liver disease, and current use of hepatotoxic medication. The absence of any viral, genetic, autoimmune, and drug-induced liver disease was confirmed by laboratory tests and medical history during the check-up. Anthropometric parameters such as weight, height and waist circumference were collected, as well as fasting metabolic biochemical parameters (glucose, HbA1c, triglycerides, total cholesterol, HDL cholesterol, and C-reactive protein).

### 2.2. Liver Steatosis Assessment

Liver steatosis was assessed by Controlled Attenuated Parameter (CAP) of transient elastography (Fibroscan 502 Touch, Echosens, Paris, France), operated by a single and experienced operator. Liver stiffness was also measured. Transient elastography was conducted according to manufacturer’s recommendations, and patients had at least four hours of fasting. The use of M or XL probe was selected according to BMI (>27 kg/m^2^). The reliability of the studies was determined by IQR < 40 and 10 valid measurements.

The diagnosis and severity of steatosis were determined according to cut-off of CAP proposed by Sirli et al. [17] as follows: non-steatosis < 263 dB/m; S1 263–282 dB/m; S2 283–295 dB/m; and S3 > 296 dB/m. Significant fibrosis was defined by liver stiffness of 7–10 kPa, whereas advanced fibrosis was defined by >10 kPa.

### 2.3. NAFLD and MAFLD Definitions

Patients were divided into two groups according to BMI: ≥25 kg/m^2^ (overweight and obesity) and <25 kg/m^2^ (lean). For overweight and obesity patients, NAFLD was defined by evidence of liver steatosis by CAP (>263 dB/m) and MAFLD was defined by evidence of steatosis by CAP (>263 dB/m). Patients with criteria of type two diabetes mellitus (DM) (previous diagnosis, fasting glucose > 126 g/dL or HbA1c > 5.6%) were included in MAFLD definition.

Lean NAFLD was defined by BMI < 25 kg/m^2^ and CAP > 263 dB/m. Lean MAFLD was defined according to consensus criteria: BMI < 25 kg/m^2^, CAP > 263 dB/m, and the presence of at least two metabolic risk abnormalities: elevation of blood pressure SBP > 130 mmHg, DBP > 85 mmHg, diagnosed arterial hypertension, impaired fasting glucose (100–125 mg/dL), abnormal postprandial glucose (140–199 mg/dL), HbA1c between 5.7 and 6.4%, triglyceride levels > 150 mg/dL, HDL cholesterol < 40 mg/dL in men, <50 mg/dL in women, patients undergoing treatment for dyslipidemia, abdominal circumference > 102 cm in men, >88 cm in women, and plasma high-sensitivity C-reactive protein levels > 2 mg/L [11].

### 2.4. Statistical Analysis

Continuous variables were described by mean and standard deviation, whereas categorical data were presented as numbers and percentages. Comparison of prevalence in each definition and BMI cut-off were analyzed by Fisher’s exact test, and effect size was evaluated by Cohen’s *d*/*w* test according to characteristics of variables. Multivariate analysis by logistic regression included variables with a *p* value < 0.1 in the univariate analysis. Odds ratios (ORs) and 95% confidence intervals (CIs) were calculated for each covariate, and a *p* value < 0.05 was accepted as significant. All statistical analysis were performed using statistical SPSS/Mac version 26.0 software (SPSS, Chicago, IL, USA).

## 3. Results

Medical records for 4271 patients were collected. After applying selection criteria, 3847 patients were included, with 59.4% (*n* = 2287) males with a mean age of 50 ± 11 years and BMI of 26.4 ± 4 kg/m^2^. The prevalence of DM was 6.2% (*n* = 240), liver steatosis was diagnosed in 46% (*n* = 1769) majorly S3 25.5% (*n* = 982). Mean of liver stiffness was 4.2 ± 1.5 kPa, and significant fibrosis was present in 0.9% (*n* = 33), and 0.6% (*n* = 23) presented advanced fibrosis. The characteristics of all patients are presented in Table 1.

Overall, 2351 patients (61.1%) had BMI ≥ 25 kg/m^2^. In these patients, the prevalence of DM was 9% (*n* = 211) and the mean of CAP was 281.4 ± 51.5 dB/m. Most patients presented two metabolic abnormalities (24%, *n* = 580), where elevated waist circumference (48.6%, *n* = 1142) and low HDL levels (44.6%, *n* = 1049) were the most common. According to NAFLD criteria, the prevalence of liver steatosis was 63.6% (*n* = 1495), and 59% (*n* = 886) corresponded to S3. When MAFLD criteria was applied, the prevalence was 65.3% (*n* = 1536) and S3 was determined in 57.7% (*n* = 886). There were no statistical differences between steatosis stages in both criteria (Figure 1); the effect size by Cohen’s W was 0.013. The prevalence of significant and advanced fibrosis was similar in the two criteria (less than 2%). The characteristics of each definition are presented in Table 2.

On the other hand, 1496 patients were lean (BMI < 25 kg/m^2^), and 54.5% (*n* = 814) were female. The prevalence of DM in lean patients was 1.9% (*n* = 28), mean of CAP was 222.8 ± 44.2 dB/m, and mean of age was 48.7 ± 10.2 years. In general, metabolic parameters were normal; however, 32.5% (*n* = 490) of patients presented one metabolic abnormality and the most common was low HDL levels (25.4%, *n* = 379). Significant fibrosis was present in 0.3% (*n* = 5). Regarding lean NAFLD criteria, the prevalence was 18.3% (*n* = 273); meanwhile, according to lean MAFLD criteria, the prevalence was lower (7.9%, *n* = 118), with the Cohen’s W test indicating a large effect size (0.87) in difference of prevalence. In lean NAFLD criteria, 45% (*n* = 123) of patients had S1. According to lean MAFLD criteria, S3 was higher, at 39.8% (*n* = 47) in these patients. However, stages of steatosis did not show significant differences (Figure 1). Significant or advanced fibrosis was not found in lean NAFLD and MAFLD patients. Since lean MAFLD criteria includes metabolic abnormalities, we observed that serum lipids impairments were the most common in 62.7% (*n* = 74) of the cases. The characteristics of each definition are presented in Table 3.

In patients with BMI < 25 kg/m^2^ (lean), univariate and multivariate analyses were performed to identify independent metabolic factors associated with steatosis. Age, biochemical, and anthropometrical data were dichotomized according to percentile 75 and 25 for HDL levels. In univariate analysis, age, anthropometrical, and biochemical data, except CRP, were associated to liver steatosis; in multivariate analysis, age higher than 54 years (OR 1.4 CI95% 1.0–1.9), BMI higher than 24 kg/m2 (OR 1.6 CI95% 1.1–2.2), and waist circumference >89 cm (OR 2.0 CI95% 1.4–2.8) showed an independent association with the presence of steatosis. According to biochemical data, glucose (OR 1.8 CI95% 1.3–2.4), triglycerides (OR 2.6 CI95% 1.9–3.5), and HDL cholesterol (OR 1.5 CI95% 1.1–2.0) showed a significant independent association (Table 4).

## 4. Discussion

The new MAFLD term not only involves a change of name but also a change in disease definition, in which metabolic risk factors play an important role. As a result, some patients who were previously diagnosed with NAFLD or lean NAFLD may not fulfil the MAFLD or lean MAFLD criteria [12]. Due to the recent coinage of the term, the impact in epidemiology is unknown. In this study, liver steatosis prevalence was higher than reported worldwide. This is an expected result because our population is Latin-American, with higher risk of liver steatosis related to genetic and obesity characteristics [18]. However, the prevalence in patients with BMI ≥ 25 kg/m^2^ did not show significant differences when both terms were applied (63% vs. 65%). Therefore, the new criteria do not seem to modify the inclusion of patients in this scenario. In a similar study, Lin et al. [19] also observed that prevalence is similar in both definitions, however those patients classified as MAFLD showed more metabolic comorbidities. In this study, liver steatosis was determinate by abdominal ultrasound and serum markers, both methods with lower diagnostic accuracy than CAP, and they did not evaluate the difference of prevalence in lean patients.

In contrast, when MAFLD criteria were applied in patients with normal BMI, prevalence changed, being higher in lean NAFLD (18.3%) than in lean MAFLD (7.9%). As for grade of steatosis, differences were not shown in the distribution. This is probably related to sample size. However, it is necessary to keep in mind that lean MAFLD patients have metabolic abnormalities (majorly high triglycerides, glucose impairments, and low HDL) and they are considered a high cardiovascular risk population, with a high predisposition to liver disease progression. We found higher prevalence than a meta-analysis by Lu et al. [20], which reported a 9.7% prevalence of liver steatosis applying NAFLD criteria. Once again, this difference is expected since prevalence in our region has been reported up to 70%. Similar differences in prevalence have been observed in lean NAFLD. However, previous studies came from Asiatic and Caucasian populations with different cut-off points for BMI; notwithstanding, younger age stands out in patients with liver steatosis and normal BMI. Evaluation of liver steatosis patients in lean patients is one of the strengths of our study. In results of univariate and multivariate analysis, we observed that metabolic alterations have strong associations with liver steatosis. Despite other proposed pathophysiology mechanisms, this group of patients should be considered as high risk, especially because liver steatosis and metabolic screening are not routine in these patients.

In a recent study, Wong et al. [16] retrospectively applied both criteria in 992 patients with steatosis, measured by proton-magnetic resonance spectroscopy. They did not observe differences in prevalence, but MAFLD criteria had a lower impact in incidence: nearly 25% of patients with liver steatosis but initially without MAFLD. This could be explained with decreased metabolic abnormalities in patients without steatosis, excluding them from the MAFLD criteria, even with increased liver fat. Although MAFLD criteria are not more discriminatory than NAFLD, when identifying patients with significant liver disease, they observed a statistical, but not clinically significant, increase of liver stiffness in patients with MAFLD, but not with NAFLD. Similar to our results, they did not find significative differences in liver stiffness in patients with BMI < 25 kg/m^2^. When patients were classified with MAFLD criteria, Kim et al. [21] observed a higher risk of cardiovascular mortality (HR 1.17 95% CI 1.04–1.32) in a study with more than 7000 patients. These results emphasize the importance of early detection of MAFLD patients.

Regarding lean patients, our results are similar to the study performed by Cheng et al. [22], with a prevalence of 16.5% in 880 patients, taking into account that the cut-off for BMI to define lean patients is lower and the prevalence is higher in the female gender in that study. However, these results are consistent with previous evidence, and, with our results insofar as patients with lean MAFLD are characterized by being older, with high levels of triglycerides and waist circumference. In Cheng et al.’s study, body composition was measured, finding that lean MAFLD patients have a different body composition (lower fat mass and corporal water) independently of BMI. However, this body composition pattern is not different from lean patients without liver steatosis. Body composition analysis deserves special interest in order to identify its possible prognostic implications with metabolic abnormalities in each patient.

In a post-hoc analysis, we identified that patients with BMI < 25 kg/m^2^ with metabolic abnormalities showed a higher risk of NAFLD/MAFLD (OR 1.4 to 2.6, Table 4). This result highlights that if NAFLD criteria is applied in these patients, 70% of lean patients could have metabolic risks that could be overlooked, impacting the early detection and progression of liver disease and/or metabolic syndrome, which have a direct relationship. In multivariate analysis, we observed that the independent risk factors for liver steatosis in lean patients are age, triglycerides, HDL, and waist circumference. Cheng et al. [22] observed similar results in 56 lean subjects; elder age was associated with lean MAFLD and these patients showed higher waist circumference. In 2021, Alam et al. [13] confirmed these associations in a meta-analysis of 22 studies in patients with BMI < 25 kg/m^2^.

The results of application of both new definitions, in a determined population, could be evaluated based on size effects, which are not dependent on sample size. According to our results, metabolic variables such as DM and HBP showed medium and large effect sizes in both lean and overweight/obese patients. Specifically in lean patients, it seems to be that new definition enables the detection of patients with higher metabolic abnormalities (biochemical, clinical, and anthropometric) that have been associated with higher cardiovascular risk and liver disease progression (steatohepatitis and fibrosis), even in the absence of overweight.

As far as we know, this is the first epidemiologic study in a Mexican/Latin-American population with both criteria for liver steatosis that includes lean patients. We evaluated liver steatosis and fibrosis by transient elastography which, along with spectroscopy, is considered one of the best diagnostic methods for evaluation of liver fat and fibrosis. Although there is no formal consensus for cut-off points for CAP, we evaluated our patients with a cut-off >263 dB/m, based on the best diagnostic accuracy for similar and homogeneous population [17]. Homeostasis model assessment of insulin resistance score is one of the metabolic abnormalities considered in lean MAFLD definition; unfortunately, this score was not available for our patients. On the other hand, the population for this study was selected in a check-up unit—this could represent a selection bias for the extrapolation of results. Another limitation was that liver steatosis was not measured by biopsy, which is a gold standard. However, transient elastography is considered an equivalent non-invasive diagnostic method nowadays.

Even though the new criteria are not universally accepted yet, they could include patients at a younger age and with higher metabolic and cardiovascular risks. This may not have an impact on prevalence, but on the increase of incidence. A previous study [21] showed that patients included in the new criteria presented significant fibrosis. The new criteria could have a global impact in the long term, starting early therapies in this population with a high risk of cardiovascular and hepatic mortality.

Evidence of new definitions is still uncertain. The studies are heterogeneous in order of methodology, selection criteria, diagnostic methods, and outcomes. However, conclusions are consistent with a better selection of metabolic unhealthy patients with MAFLD and lean MAFLD criteria. These patients have more cardiovascular risk and a higher risk of liver fibrosis and disease progression [23]. The prevalence of liver steatosis in Western and specifically Latin-American populations is close to 60%, representing an important impact on health systems over the next decade and highlighting early detection and treatment for liver steatosis for all patients but also focused on those who could be underestimated. 

In previous years, we have focused on liver steatosis and underestimated metabolic comorbidities. MAFLD criteria could be an additional and strict diagnostic tool for the early detection of high risk of metabolic and hepatic abnormalities in patients, with the purpose of starting timely lifestyle and/or pharmacological therapies, thus avoiding the progression of metabolic and liver diseases.

## 5. Conclusions

The application of NAFLD/MAFLD criteria did not show prevalence differences in patients with BMI ≥ 25 kg/m^2^. With MAFLD criteria, prevalence is lower in lean patients, but it identifies high risks of progression of liver disease from steatosis, according to metabolic and anthropometric abnormalities.

## Figures and Tables

**Figure 1 ijerph-19-12221-f001:**
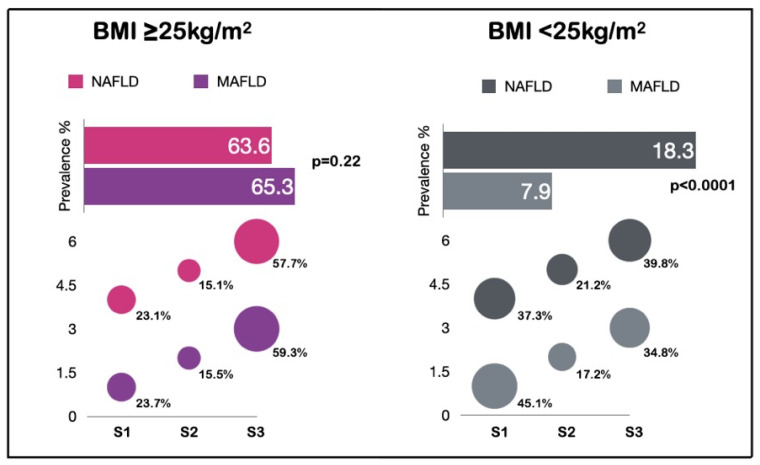
Differences on prevalence according to NAFLD and MAFLD criteria. BMI: body mass index; NAFLD: non-alcoholic fatty liver disease; MAFLD: metabolic dysfunction-associated fatty liver disease; S1 263–282 dB/m; S2 283–295 dB/m; S3 > 296 dB/m.

**Table 1 ijerph-19-12221-t001:** Clinical and demographic characteristics of all patients (*n* = 3847).

Characteristic	% (*n*)/μ ± SD
Male	59.4 (2287)
Age (years)	50±11
DM	6.2 (240)
HBP	13.5 (521)
BMI (kg/m^2^)	26.4 ± 4.0
BMI ≥ 25 kg/m^2^	61.1 (2351)
Glucose (g/dL)	94.7 ± 22.4
HbA1c (%)	5.4 ± 0.7
SBP (mmHg)	112.4 ± 11.15.2
DBP (mmHg)	73.4 ± 9.4
Triglycerides (mg/dL)	133.1 ± 85.3
HDL (mg/dL)	49.2 ± 16.4
CRP (mg/L)	2.6 ± 7.5
WC (cm)	92.3 ± 11.6
CAP (dB/m)	258 ± 56.5
Steatosis by CAP > 263 db/m	46 (1769)
S1	12.4 (478)
S2	7.3 (279)
S3	25.5 (982)
kPa	4.2 ± 1.5
Significant Fibrosis 7–10 kPa	0.9 (33)
Advanced fibrosis > 10 kPa	0.6 (23)

DM: diabetes mellitus; HBP: high blood pressure; BMI: body mass index; HbA1C: glycosylated hemoglobin; SBP: systolic blood pressure; DBP: diastolic blood pressure; HDL: high density lipoprotein; CRP: C reactive protein; WC: waist circumference; CAP: controlled attenuation parameter.

**Table 2 ijerph-19-12221-t002:** Differences in clinical and demographic characteristics of NAFLD and MAFLD criteria in patients with BMI ≥ 25 kg/m^2^ (*n* = 2351).

Characteristic	NAFLD (*n* = 1495) % (*n*)/μ ± SD	MAFLD (*n* = 1536) % (*n*)/μ ± SD	*p*	Effect Size (Cohen’s d/w)	
Male	71.6 (1071)	71.4 (1097)	0.90	0.28	Small
Age (years)	50.8 ± 13.1	51.1 ± 13.1	0.52	−0.01	Null
DM	11.4 (170)	13.7 (211)	**0.05**	0.87	Large
HBP	20.7 (310)	21.5 (330)	0.62	0.78	Medium
BMI (kg/m^2^)	29.5 ± 3.5	29.5 ± 3.5	1.00	−0.70	Null
Glucose (g/dL)	100.2 ± 25.8	100.7 ± 26.3	0.59	−0.31	Null
HbA1c (%)	5.6 ± 0.9	5.6 ± 0.9	1.00	−0.28	Null
SBP (mmHg)	117.3 ± 15.8	117.4 ± 15.8	0.86	−0.27	Null
DBP (mmHg)	76.7 ± 9.3	76.6 ± 9.4	0.76	−0.37	Null
Triglycerides (mg/dL)	165.8 ± 99	165.2 ± 98.4	0.86	−0.37	Null
HDL (mg/dL)	44.3 ± 18	44.3 ± 17.9	1.00	0.22	Small
CRP (mg/L)	3.2 ± 5.9	3.3 ± 6.8	0.66	−0.10	Null
WC (cm)	100.7 ± 9.7	100.6 ± 9.7	0.77	−0.74	Null
CAP (dB/m)	311.7 ± 35.7	309.5 ± 38	0.10	−2.44	Null
S1	23.7 (355)	23.1 (355)	0.69	0.76	Medium
S2	15.5 (232)	15.1 (232)	0.76	0.84	Large
S3	59.3 (886)	57.7 (886)	0.39	0.41	Small
kPa	4.6 ± 1.9	4.6 ± 1.9	1.00	−0.23	Null
Significant fibrosis 7–10 kPa	1.6 (24)	1.6 (25)	1.00	0.98	Large
Advanced fibrosis >10 kPa	1.3 (20)	1.4 (22)	0.87	0.98	Large

DM: diabetes mellitus; HBP: high blood pressure; BMI: body mass index; HbA1C: glycosylated hemoglobin; SBP: systolic blood pressure; DBP: diastolic blood pressure; HDL: high density lipoprotein; CRP: C reactive protein; WC: waist circumference; CAP: controlled attenuation parameter.

**Table 3 ijerph-19-12221-t003:** Differences in clinical and demographic characteristics of NAFLD and MAFLD criteria in patients with BMI < 25 kg/m^2^.

Characteristic	Lean NAFLD (*n*= 273) % (*n*)/μ ± SD	Lean MAFLD (*n* = 118) % (*n*)/μ ± SD	*p*	Effect Size (Cohen’s d/w)	
Male	56.8 (155)	52.5 (62)	0.44	0.92	Large
Age (years)	50.7 ± 9.9	52.7 ± 10.5	0.07	−0.06	Null
DM	4.8 (13)	10.2 (12)	0.06	0.99	Large
HBP	10.3 (28)	17.8 (21)	**0.04**	0.98	Large
BMI (kg/m^2^)	23.4 ± 1.8	23.3 ± 2.4	0.65	−0.69	Null
Glucose (g/dL)	94.2 ± 16.5	98.9 ± 22.2	**0.02**	−0.40	Null
HbA1c (%)	5.4 ± 0.6	5.5 ± 0.7	0.15	−0.39	Null
SBP (mmHg)	110.4 ± 14.1	114.3 ± 13.8	**0.01**	−0.31	Null
DBP (mmHg)	72.4 ± 9.2	74.7 ± 9.6	**0.02**	−0.36	Null
Triglycerides (mg/dL)	138.8 ± 81.1	184.2 ± 95.8	**<0.0001**	−0.37	Null
HDL (mg/dL)	48.1 ± 12.7	43.2 ± 12.6	**0.0005**	0.23	Small
CRP	2.8 ± 18.3	5.2 ± 27.7	0.31	−0.14	Null
WC (cm)	87.7 ± 7.0	88.7 ± 7.9	0.21	−0.75	Null
CAP (dB/m)	291.9 ± 26.5	296.3 ± 27.9	0.13	−2.24	Null
S1	45.1 (123)	37.3 (44)	0.18	0.94	Large
S2	17.2 (47)	21.2 (25)	0.39	0.97	Large
S3	34.8 (95)	39.8 (47)	0.36	0.95	Large
kPa	4.0 ± 0.8	4.1 ± 0.8	0.257	−0.25	Null
Glucose abnormalities	26.4 (72)	44.9 (53)	**0.004**	0.95	Large
HBP abnormalities	17.2 (47)	31.4 (37)	**0.003**	0.97	Large
Triglycerides >150	32.6 (89)	62.7 (74)	**0.0001**	0.94	Large
HDL abnormalities	35.5 (97)	62.7 (74)	**0.0001**	0.94	Large
CRP > 2.0	23.8 (65)	44.9 (53)	**0.0001**	0.96	Large
WC abnormalities	10.3 (28)	18.6 (22)	**0.03**	0.98	Large

DM: diabetes mellitus; HBP: high blood pressure; BMI: body mass index; HbA1C: glycosylated hemoglobin; SBP: systolic blood pressure; DBP: diastolic blood pressure; HDL: high density lipoprotein; CRP: C reactive protein; WC: waist circumference; CAP: controlled attenuation parameter.

**Table 4 ijerph-19-12221-t004:** Univariate and multivariate analysis for steatosis in patients with BMI < 25 kg/m^2^.

Characteristic	Univariate		Multivariate	
	OR (CI 95%)	*p*	OR (CI 95%)	*p*
Age > 54 years	1.65 (1.24–2.20)	**0.001**	1.42 (1.02–1.97)	**0.036**
BMI > 24 kg/m^2^	2.54 (1.92–3.36)	**≤0.0001**	1.63 (1.19–2.24)	**0.002**
SBP > 117 mmHg	1.70 (1.27–2.27)	**≤0.0001**	1.04 (0.72–1.50)	0.801
DBP > 76 mmHg	1.79 (1.34–2.38)	**≤0.0001**	1.20 (0.84–1.71)	0.307
Fasting glucose > 94 mg/dL	2.58 (1.94–3.42)	**≤0.0001**	1.80 (1.30–2.48)	**≤0.0001**
Triglycerides > 124 mg/dL	3.77 (2.86–4.97)	**≤0.0001**	2.63 (1.94–3.58)	**≤0.0001**
HDL < 44 mg/dL	2.57 (1.95–3.37)	**≤0.0001**	1.52 (1.12–2.08)	**0.007**
CRP > 1.80 mg/dL	1.28 (0.95–1.73)	0.099	0.97 (0.70–1.34)	0.869
HbA1c > 5.5%	1.53 (1.13–2.08)	**0.006**	0.87 (0.61–1.25)	0.476
WC > 89 cm	3.45 (2.59–4.58)	**≤0.0001**	2.04 (1.47–2.83)	**≤0.0001**

BMI: body mass index; HbA1C: glycosylated hemoglobin; SBP: systolic blood pressure; DBP: diastolic blood pressure; HDL: high density lipoprotein; CRP: C reactive protein; WC: waist circumference.

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
