# Peer review of "Impact on Prevalence of the Application of NAFLD/MAFLD Criteria in Overweight and Normal Weight Patients"

_ijerph, 2022, doi:10.3390/ijerph191912221_

Round 1

Reviewer 1 Report

The manuscript presents an interesting topic regarding the application of NAFLD and MAFLD criteria in overweight and normal-weight patients. Nevertheless, as a reviewer, I must point out some deficiencies in the text. Please explain the following aspects in detail.

- Lines 26 and 27: the cut-off points for obese and lean patients should be mentioned.

- Lines 35: In the introduction section, I propose including more information about lifestyle factors that predispose to the development of NAFLD/MAFLD, such as a type of diet, a type of microbiome disorder, and the region where NAFLD/MAFLD is most prevalent. What about the treatment process and health prognosis? etc. 

- Line 63: Inclusion criteria for the study should be described (e.g., age, ethnicity, gender, clinical symptoms, and others).

- Line 69: Which anthropometric and biochemical parameters were marked?

- Please pay special attention to punctuation in the entire manuscript (kg/m2 - „2” should be in a square; the reference number in brackets should be before of the dot at the end of the sentence; the BMI value indicating excessive body weight should be ≥ 25.0 kg/m2 (not >25.0 kg/m2).

- Line 189 and 196: references are required.

- Lines 191, 204, 210: and entire manuscript: the reference number in brackets should be directly behind the name of the authors, not at the end of the sentence.

The references are very poor. The authors should develop the Introduction section and include in the discussion the works of other authors regarding a similar topic, e.g.,

- Kuchay MS, Choudhary NS, Mishra SK. Pathophysiological mechanisms underlying MAFLD. Diabetes Metab Syndr. 2020;14(6):1875-1887. doi:10.1016/j.dsx.2020.09.026

- Lin S, Huang J, Wang M, et al. Comparison of MAFLD and NAFLD diagnostic criteria in real world. Liver Int. 2020;40(9):2082-2089. doi:10.1111/liv.14548

- Fouad Y, Elwakil R, Elsahhar M, et al. The NAFLD-MAFLD debate: Eminence vs evidence. Liver Int. 2021;41(2):255-260. doi:10.1111/liv.14739

A suggestion for the next scientific and epidemiological investigations: for the next work, the Authors could plan to include body composition analysis, which would allow to obtain more comprehensive results.

Author Response

Dear reviewer,

We read all your comments regarding the paper entitled: Impact on prevalence of the application of NAFLD/MAFLD criteria in overweight and normal weight patients. All comments were very useful and undoubtedly improve significantly the quality of the manuscript.

- Lines 26 and 27: the cut-off points for obese and lean patients should be mentioned.

R= Thank you for your observation, cut-off points for overweight/obese and lean patients were included.

- Lines 35: In the introduction section, I propose including more information about lifestyle factors that predispose to the development of NAFLD/MAFLD, such as a type of diet, a type of microbiome disorder, and the region where NAFLD/MAFLD is most prevalent. What about the treatment process and health prognosis? etc. 

R= Thank you for your comment, introduction section was improved including more information about epidemiology, physio pathological mechanisms and actual treatment recommendations. (Lines 37 to 66)

- Line 63: Inclusion criteria for the study should be described (e.g., age, ethnicity, gender, clinical symptoms, and others).

R= Thank you for your observation, inclusion criteria were included (Lines 100-101)

- Line 69: Which anthropometric and biochemical parameters were marked?

R= Anthropometric and biochemical parameters were specified in lines 105 to 107

- Please pay special attention to punctuation in the entire manuscript (kg/m2 - „2” should be in a square; the reference number in brackets should be before of the dot at the end of the sentence; the BMI value indicating excessive body weight should be ≥ 25.0 kg/m(not >25.0 kg/m2).

R= Thank you very much for your observation, all units of BMI were corrected as well as reference’s punctuation, also we change “>” for “≥” in all cases that refers to overweight or obesity.

- Line 189 and 196: references are required.

R= Thank you for your comment, data mentioned in these lines refers to our results. (Now line 244)

- Lines 191, 204, 210: and entire manuscript: the reference number in brackets should be directly behind the name of the authors, not at the end of the sentence.

R= Thank you for your observation, all punctuation was corrected

The references are very poor. The authors should develop the Introduction section and include in the discussion the works of other authors regarding a similar topic, e.g.,

- Kuchay MS, Choudhary NS, Mishra SK. Pathophysiological mechanisms underlying MAFLD. Diabetes Metab Syndr. 2020;14(6):1875-1887. doi:10.1016/j.dsx.2020.09.026

- Lin S, Huang J, Wang M, et al. Comparison of MAFLD and NAFLD diagnostic criteria in real world. Liver Int. 2020;40(9):2082-2089. doi:10.1111/liv.14548

- Fouad Y, Elwakil R, Elsahhar M, et al. The NAFLD-MAFLD debate: Eminence vs evidence. Liver Int. 2021;41(2):255-260. doi:10.1111/liv.14739

R= Thank you very much for your comment. Several references were included in entire manuscript improving introduction and discussion sections. Your suggestion of references was very useful, we included this studies in our manuscript (References 7, 19 and 24)

A suggestion for the next scientific and epidemiological investigations: for the next work, the Authors could plan to include body composition analysis, which would allow to obtain more comprehensive results.

R= Thank you for your comment. Actually, we are working in a prospective study with obese and lean MAFLD patients, in which we evaluate body composition, we hope to published our results as soon as possible.

All the authors appreciate the reviewer’s time invested in our manuscript. If you need further information, please contact me

Kind regards,

 Iván López-Mendez. Transplants and Hepatology Unit. Medica Sur Clinic & Foundation. Puente de Piedra 150 Toriello Guerra, Tlalpan. ZC 14050. Mexico City, Mexico. Phone: 5424-6850. E-mail: [email protected]

Reviewer 2 Report

This study aims to identify changes in prevalence and metabolic profiles using NAFLD and MAFLD definitions in both lean and overweight Mexican/Latin-American patients. In the study, it is found that overweight patients are not affected differently by NAFLD versus MAFLD in terms of prevalence. MAFLD criteria is found to include patients at a younger age and with higher metabolic and cardiovascular risks.

The study is well-conducted. The results represent a large sample of patients. Some issues should be addressed to improve the content of the manuscript:

-The Patient population section should include information on the number of patients recruited for the study.

-The title of table 4 should be corrected.

-Cohen's d or w analysis would be highly recommended to estimate the significance of the differences found in Tables 2 and 3.

-Are the results of this study really useful? The authors are encouraged to expand on this further in the discussion.

Author Response

Dear reviewer,

We read all your comments regarding the paper entitled: Impact on prevalence of the application of NAFLD/MAFLD criteria in overweight and normal weight patients. All comments were very useful and undoubtedly improve significantly the quality of the manuscript.

-The Patient population section should include information on the number of patients recruited for the study.

R= Thank you for your comment. Total of medical records collected (4271) for the study was included in Results section (line 145)

-The title of table 4 should be corrected.

R= Thank you for your observation, Title of table 4 was corrected. 

-Cohen's d or w analysis would be highly recommended to estimate the significance of the differences found in Tables 2 and 3.

R= Thank you very much for your comment and suggestion. Choen’s d and w was calculated and included for results of Tables 2 and 3, also we analyzed the effect size for prevalence differences (lines 165 and 185)

-Are the results of this study really useful? The authors are encouraged to expand on this further in the discussion.

R= Thank you for your comment. We improved the Discussion section highlighting the strengths of our study such as analysis of lean NAFLD/MAFLD patients and the importance of early detection as high-risk patients.

All the authors appreciate the reviewer’s time invested in our manuscript. If you need further information, please contact me

Kind regards,

 Iván López-Mendez. Transplants and Hepatology Unit. Medica Sur Clinic & Foundation. Puente de Piedra 150 Toriello Guerra, Tlalpan. ZC 14050. Mexico City, Mexico. Phone: 5424-6850. E-mail: [email protected]

Round 2

Reviewer 1 Report

Much thanks to the authors for incorporating major recommendations from the review, with the paper better suitable for publication.

Let me limit to one further comment of content:

Line 101: „old apparently healthy” - I suggest to changing this phrase to: "without diagnosed chronic diseases" or completely removing this sentence, due to lines 103-104 where exclusion diseases were mentioned.

Author Response

Thank you, we decided to remove "apparently healthy". 

Once again, thank you very much for your suggestions and comments 

Reviewer 2 Report

Effect sizes should be categorised as null, small, medium or large (Cohen, 1988, "Statistical power analysis for the behavioural sciences"). This should be included in the tables. And the discussion should focus especially on results with at least medium effect sizes. Differences with small effect sizes could perhaps also be discussed, but this would be a limitation.

Author Response

Dear reviewer, 

Thank you for your suggestion. Size effect (null, small, medium, large) was included in Tables 2 and 3. In discussion section we also included this results (lines 296-303)

Once again thank you for all your suggestions and comments.